# Quantification and Evaluation of Water Requirements of Oil Palm Cultivation for Different Climate Change Scenarios in the Central Pacific of Costa Rica Using APSIM

Fernando Watson-Hernández [1],*, Valeria Serrano-Núñez [1], Natalia Gómez-Calderón [1] and Rouverson Pereira da Silva [2]

1   School of Agricultural Engineering, Instituto Tecnológico de Costa Rica, Cartago 30101, Costa Rica
2   Department of Engineering and Mathematical Sciences, School of Agricultural and Veterinarian Sciences, São Paulo State University (Unesp), Jaboticabal 14884-900, São Paulo, Brazil
*   Correspondence: fwatson@itcr.ac.cr; Tel.: +506-8474-6643

**Abstract:** Climate change is a variation in the normal behavior of the climate. These variations and their effects will be seen in the coming years, the most imminent being anomalous fluctuations in atmospheric temperature and precipitation. This scenario is counterproductive for agricultural production. This study evaluated the effect of climate change on oil palm production for conditions in the Central Pacific of Costa Rica, in three simulation scenarios: the baseline between the years 2000 and 2019, a first climate change scenario from 2040 to 2059 (CCS1), and a second one from 2080 to 2099 (CCS2), using the modeling framework APSIM, and the necessary water requirements were established as an adaptive measure for the crop with the irrigation module. A decrease in annual precipitation of 5.55% and 7.86% and an increase in the average temperature of 1.73 °C and 3.31 °C were identified, generating a decrease in production yields of 7.86% and 37.86%, concerning the Baseline, in CCS1 and CCS2, respectively. Irrigation made it possible to adapt the available water conditions in the soil to maintain the baseline yields of the oil palm crop for the proposed climate change scenarios.

**Keywords:** irrigation; climate variability; digital agriculture; crop simulation

## 1. Introduction

Oil palm is a tropical evergreen tree; due to its high yield and relatively low production costs, it has become one of the most popular products, surpassing other vegetable oils, and one of the most profitable lands uses in humid tropical areas [1,2]. Oil palm is one of the crops with the largest planting area in Costa Rica, representing 0.15% of the GDP of the country, with the Central Pacific being where half of the palm oil mills are located [3,4], so the prediction of scenarios promotes the improvement in the logistics of the production system.

Being a tropical plant species, the oil palm requires warm conditions and high humidity to take full advantage of its photosynthetic capacity, so its optimum temperature conditions fluctuate between 24 °C and 28 °C and daily solar radiation (Rad) of 16–17 MJ m$^{-2}$, with an average maximum temperature of 30–32 °C and an average minimum of 21–24 °C, where the coldest month of the year cannot be less than 15 °C; it also needs a high rainfall, with annual precipitation between 2000 and 2500 mm, with a minimum of 100 mm per month. However, it can withstand rainfall of 4000 mm up to a limit of 5000 mm per year in well-drained soils (although the probability of disease increases). On the contrary, there have been reported cases of plantations with rainfall less than 1000 mm per year and dry periods of up to five months [5].

According to Paterson and Lima [6], optimal climatic conditions for palm cultivation will be gradually affected by 2030 and even more so by 2100, due to the effects of climate

change (CC) in tropical regions. An increase in temperature and low rainfall is estimated [7], causing oil palm production yields to gradually decrease worldwide [8].

For instance, Tani et al. [9] indicate that the female reproductive organ of oil palm can be affected by climatic variability through various reproductive processes, resulting in bunch abortion and delay of fruit bunch growth. Therefore, it is necessary to investigate how environmental conditions and biotic factors influence plant behavior.

Currently, the critical levels of water deficit in the different stages of palm development and the optimum volumes of water to be applied are not established [10], and the degree of water deficit stress is the decisive point for knowing when to irrigate. This parameter can be determined from the water dynamics in the water retention curve in the soil where the crop grows [11].

The effects of climate change on the crop can be simulated using computational tools. APSIM (Agricultural Production Systems Simulator) is a modeling framework for biophysical processes in agricultural systems and is one of the main tools for understanding the economic and environmental performance of agricultural management in the face of climate risks [12]. APSIM uses different components of an agricultural system and therefore requires information on climatic conditions, soil conditions, and management practices.

APSIM's variety of key modules has made it a better agricultural systems simulator, making it a very useful framework for evaluating changes in agricultural landscapes [13], such as the oil palm module that has been calibrated and validated for tropical conditions [14]. Furthermore, APSIM is a modeling framework with a simple, easy-to-use, intuitive interface that is flexible enough for a wide range of scenario analyses, with a good execution speed and is constantly being improved [15].

APSIM has been increasingly applied in assessing the effects of climate change on yields of crops such as maize and wheat [16–19], and oil palm [20], as well as adaptation and mitigation strategies [21].

This study is in line with the Sustainable Development Goals (SDGs), specifically number 13, which proposes actions to combat climate change and its impacts. The purpose of this study is to evaluate the effect of climate change on oil palm production in the Central Pacific region of Costa Rica using the modeling framework APSIM and to determine the necessary water requirements as an adaptive measure for the crop, providing information to the productive sector for its due preparation for the coming changes.

## 2. Materials and Methods

The research subjects were oil palm located in the Central Pacific region of Costa Rica, with a plantation area of 9445.74 hectares subdivided into a total of 186 agricultural production blocks (APBs) (Figure 1). The sample size was 19 APBs, chosen in a quasi-experimental way, considering as a discriminating variable the crop yield for 20-year data series, resulting finally in the selection of 8 representative APBs, on which climate change scenarios (CCS) were contemplated.

### 2.1. Study Area

A selection of the working APBs was made, which was subdivided into two parts: a pre-selection that consisted of using the APBs that had a planting date of 1995 or 1996 and that had less than 5% of missing data. Subsequently, the pre-selected APBs were grouped according to performance behavior, so that the groups would be as representative as possible of the performance behavior of all the APBs. For this purpose, the Silhouette method [22,23] was used in RStudio and the clustering was performed with the Computes Hierarchical Clustering and Cut the Tree method [24,25] according to the historical yield values of each APB. Once the clustering was made and the cut of each group was defined, a representative APB was randomly selected from each one.

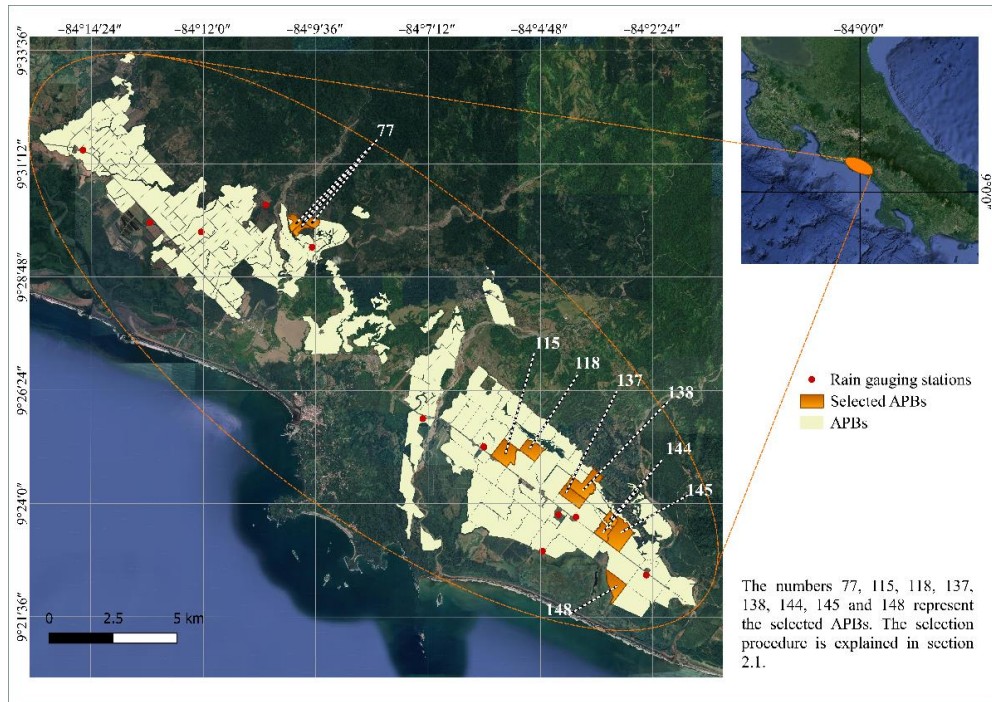

**Figure 1.** Location of oil palm plantation under study. Oil palm plantation area (yellow and orange APBs) of which eight APBs were selected (orange APBs) and the position of the rain gauging stations (red dots).

*2.2. Soil Physical Parameters*

The crop under study is located in ultisol soil, which is characterized by being deep, well-developed soil, with good structure and 1:1 clay [26].

Soil sampling was performed for each APB at four different depths up to a maximum of 90 cm, since the highest root density is located up to 60 cm soil depth [11,27], to be analyzed in the laboratory, determining the texture, bulk density (BD), soil air dry (AirDry), 15 Bar (−1500 kPa) lower limit (LL15), drained upper limit (DULL), available soil water (ASW) and soil water saturation (SAT) [28], according to the methodologies proposed by Beretta et al. [29] and Gómez-Calderón and Estrada-León [30].

*2.3. Climate Data*

Regarding climatological variables, daily data of precipitation (P), radiation (Rad), maximum (maxT), and minimum temperature (minT) in the period from 1995 to 2015 were used, in which, the accumulated annual rainfall is 4537 mm, with an average temperature range between 22 °C and 32 °C. There is a rainy season from May to November and a dry season between December and April.

Precipitation data were obtained from nearby rain gauging stations, where those with less than 20% of missing data were selected. Gaps in the data were filled by using multivariate regressions. The inverse distance weighted (IDW) method was used to obtain the precipitation at the centroid of each APB.

*2.4. Climate Change Scenarios (CCS)*

The CCS used as a basis for this study was Hidalgo et al. [31], in which the focus was on the ability of CMIP5 CGM model runs in reproducing basic climatic characteristics of Central America; the scenario used was cesm1_cam5 (model of the Euro-Mediterranean Center for Climate Change) because it was the simulation that had the best capacity to reproduce the basic characteristics of the monthly precipitation and temperature variables from a total of 107 runs, which generated a projection of monthly climate change in the

period from 1979 to 2099 for the variables of minimum temperature, maximum temperature, and monthly accumulated precipitation [32].

Since the APSIM framework requires the climate data to be on a daily scale, the CCS variables were disaggregated by stochastic methods because they have a monthly scale. For this process, the WeaGETS program [33,34] was used to obtain the CCS on a daily scale. The daily CCS was then standardized and readjusted using the mean and standard deviation of the data observed in the period 2000–2015, to correct the patterns between the trend lines and variation of these data. This scenario does not have radiation data, so two machine learning (ML) models were generated: Random Forest (RF) [35] and Neural Network (NN) [36] based on the observed data and where the radiation is in terms of maxT, minT, P, and the number of the day of the year [37]. The best model was selected using the coefficients: Nash-Sutcliffe model (NSE) (Equation (1)), the root mean square error (RMSE) (Equation (2)), the mean absolute error coefficient (MAE) (Equation (3)), and the coefficient of determination ($R^2$) (Equation (4)), where the best-qualified used to apply it in the CCSs and generate the radiation data series corresponding to the study period.

*2.5. APSIM Simulation Process*

The APSIM framework made it possible to link different components of the oil palm production system. This tool has a specific module for oil palm crops [14] that simulates the crop's growth and yield considering different climatic conditions [38].

In the APSIM modeling framework, the following modules were used: Management Palm, Meteorological, Crop Module (Oil Palm), and Soil Module (Soil Water, Irrigation). The irrigation module was only used to determine water requirements to mitigate the effects produced in the CCSs. The modules and variables used in APSIM are shown in Figure 2.

In the calibration process, models were generated for each of the selected APBs, to achieve an NSE => 0.56, and an $R^2$ => 0.62, according to the values obtained by the creators of APSIM's oil palm module [14]. This calibration was performed by adjusting soil moisture values, specifically, AirDry, LL15, DULL, and SAT. The performance evaluation of the models on each of the APBs was conducted by using the coefficients NSE (Equation (1)), RMSE (Equation (2)), MAE (Equation (3)), and $R^2$ (Equation (4)).

$$\text{NSE} = 1 - \frac{\sum_{i=1}^{n} \left( y_i^{obs} - y_i^{sim} \right)^2}{\sum_{i=1}^{n} \left( y_i^{obs} - \overline{y}^{obs} \right)^2} \tag{1}$$

$$\text{RMSE} = \sqrt{\frac{1}{n} \sum_{i=1}^{n} (y_i^{obs} - y_i^{sim})^2} \tag{2}$$

$$\text{MAE} = \frac{1}{n} \sum_{i=1}^{n} \left| y_i^{obs} - y_i^{sim} \right| \tag{3}$$

$$R^2 = \left( \frac{\sum \left[ \left( y_i^{obs} - \overline{y}^{\,obs} \right) \left( y_i^{sim} - \overline{y}^{\,sim} \right) \right]}{\sqrt{\sum \left( y_i^{obs} - \overline{y}^{\,obs} \right)^2 * \sum \left( y_i^{sim} - \overline{y}^{\,sim} \right)^2}} \right)^2 \tag{4}$$

where *n* is the amount of data available for the study, $y_i^{obs}$ corresponds to the observed data, $\overline{y}^{\,obs}$ is the average of the observed data obtained for the interval under study, $y_i^{sim}$ is the simulated data, and $\overline{y}^{\,sim}$ is the average of the simulated data.

With the models calibrated for each APB, three types of simulations were performed: the base, between the years 2000 and 2019 (baseline), a first climate change scenario from 2040 to 2059 (CCS1), and a second end-of-century scenario from 2080 to 2099 (CCS2). Baseline yields were compared against the two climate change scenarios to determine if there are variations in crop yield and water demand between them.

To determine what would be irrigation requirements to maintain the yields obtained in the baseline (2000–2019), the APSIM irrigation module was activated, for which a root depth of 0.75 cm, an irrigation application efficiency of 80%, and, a varied exhaustion percentage was assumed, so that the crop yield in the evaluated CCSs would reach the crop yield observed in the baseline. Subsequently, the total volumes of water required were quantified. This value represents the amount of water that should be applied monthly in each scenario. Figure 3 shows schematically the methodology used in the study.

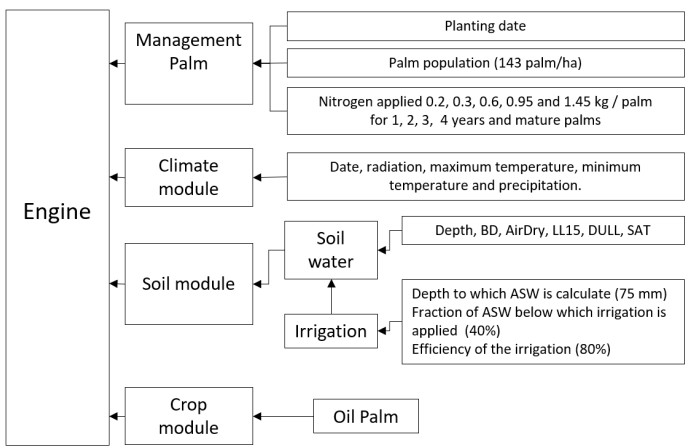

**Figure 2.** Flow diagram of the different modules of the APSIM simulation process.

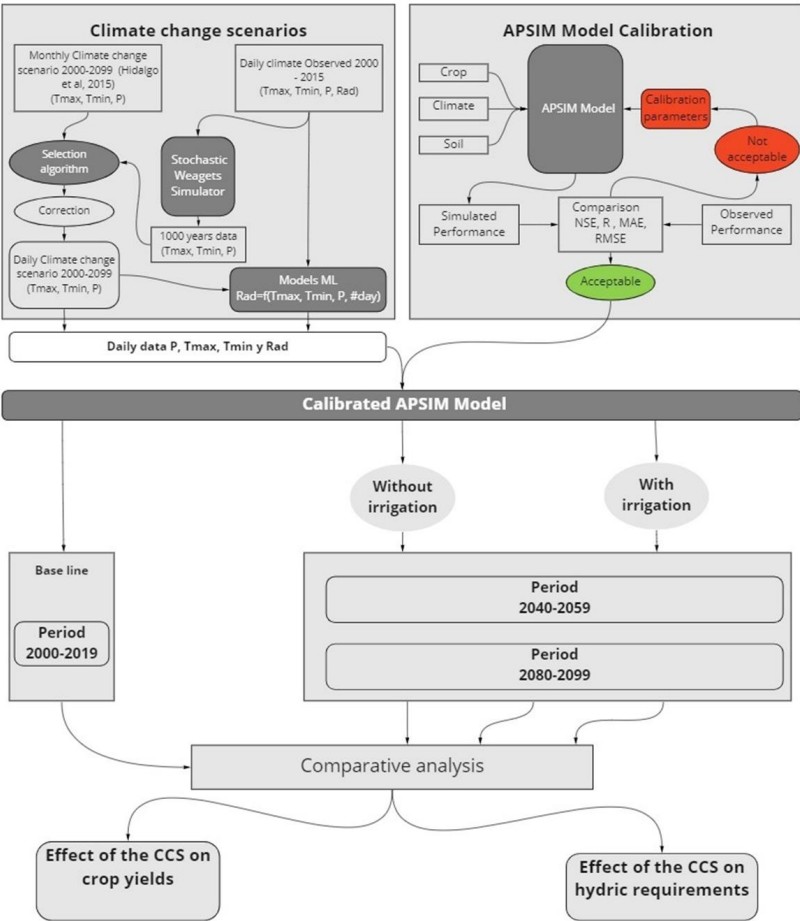

**Figure 3.** Methodological diagram of the data analysis.

## 3. Results

The results of the APBs selection parameters, soil physical parameters, climatological variables, climate change scenarios, calibration of the APSIM model, effect of climate change on oil palm production, and water requirements of the plantation in the different scenarios under study are shown below.

### 3.1. Selection of Study APBs

The number of sub-groupings of APBs with homogeneous behavior was determined using the Silhouette method according to the historical yield values of each APB, obtaining that the best possible groupings consisted of two or eight clusters. Thus, the clustering of eight was selected to maximize the representation of the APB's performance behavior. The APBs selected for the study are identified by the following numbers: 77, 115, 118, 137, 138, 144, 145, and 148 (Figure 4), where APB 144 represents the curve with the best production yield, while APB 77 is one of the curves with the lowest production yield (Figure 5).

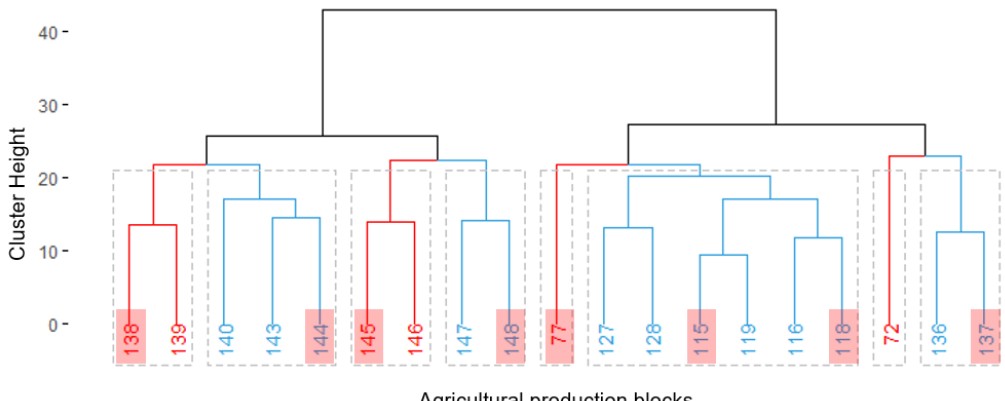

**Figure 4.** Clusters of the oil palm plantation according to monthly yield. There are eight clusters in total (dotted gray squares), the lines of different colors (red and light blue) separate the clusters, and the APBs selected in each cluster are highlighted in red.

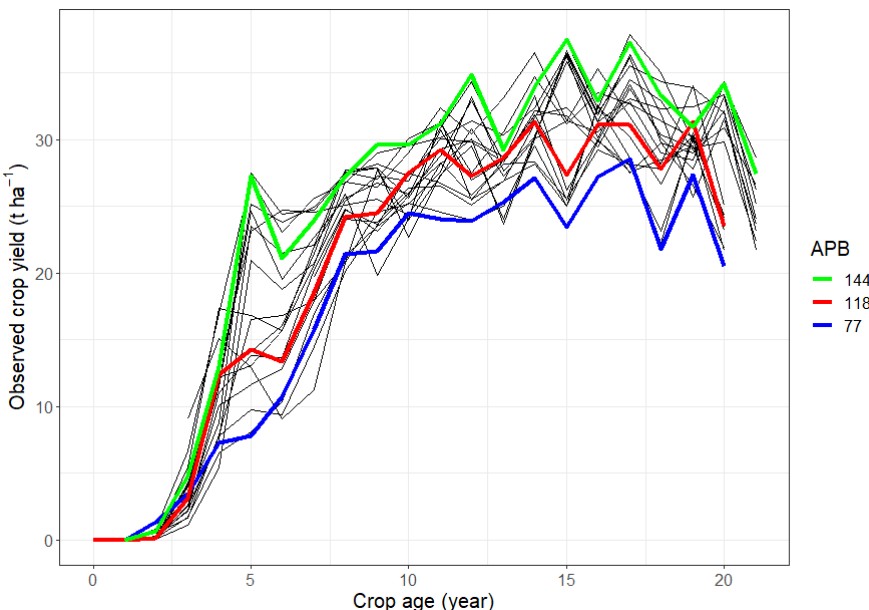

**Figure 5.** Yield behavior according to APBs plant age. Each black line represents the yield behavior of each APB. APB 144 (green line) is the best yielding APB, APB 118 (red line) represents the average yield of all APBs, and APB 77 (blue line) is the lowest yielding APB.

APB 72 was preliminarily selected but was replaced by APB 118 because it represents the average yield behavior of the population analyzed better (Figure 5). APB 72 was also excluded for missing data and an anomalous drop in its productive yield.

### 3.2. Soil Physical Parameters

The predominant textures in the selected APBs are sandy clay loam and clay loam, except for APB 148, which is clay, and APB 145, which showed loam textures in it. On the other hand, the bulk density ranges between 1.2 and 1.66 g cm$^{-3}$, with a mean and standard deviation of 1.28 g cm$^{-3}$ and 0.12 g cm$^{-3}$, respectively (Table 1).

**Table 1.** Soil texture and bulk density of selected APBs.

| APB | Depth (cm) | BD (g cm$^{-3}$) | Texture |
|---|---|---|---|
| 77 | 0.00 | 1.59 | Clay loam |
| | 25.00 | 1.35 | Sandy clay loam |
| | 50.00 | 1.22 | Sandy clay loam |
| | 90.00 | 1.22 | Sandy clay loam |
| 115 | 0.00 | 1.20 | Sandy clay loam |
| | 30.00 | 1.20 | Sandy clay loam |
| | 60.00 | 1.30 | Sandy clay loam |
| | 70.00 | 1.30 | Sandy clay loam |
| 118 | 0.00 | 1.32 | Sandy clay loam |
| | 30.00 | 1.33 | Sandy clay loam |
| | 60.00 | 1.30 | Sandy clay loam |
| | 60.00 | 1.35 | Sandy clay loam |
| | 60.00 | 1.40 | Sandy clay loam |
| 137 | 0.00 | 1.66 | Clay loam |
| | 20.00 | 1.50 | Sandy clay loam |
| | 35.00 | 1.30 | Clay loam |
| | 50.00 | 1.30 | Clay loam |
| | 80.00 | 1.40 | Clay loam |
| 138 | 0.00 | 1.03 | Clay loam |
| | 20.00 | 1.03 | Sandy clay loam |
| | 35.00 | 1.20 | Clay loam |
| | 50.00 | 1.20 | Clay loam |
| | 80.00 | 1.20 | Clay loam |
| 144 | 0.00 | 1.25 | Clay loam |
| | 30.00 | 1.22 | Sandy clay loam |
| | 50.00 | 1.22 | Sandy clay loam |
| | 87.50 | 1.22 | Sandy clay loam |
| 145 | 0.00 | 1.30 | Sandy clay loam |
| | 30.00 | 1.22 | Loam |
| | 50.00 | 1.22 | Loam |
| | 87.50 | 1.22 | Loam |
| 148 | 0.00 | 1.34 | Clay |
| | 30.00 | 1.15 | Clay |
| | 50.00 | 1.20 | Clay |
| | 87.50 | 1.30 | Clay |

APBs 138, 144, 145, and 148 had a greater range of available soil moisture, on average LL15 = 0.12 mm mm$^{-1}$ and DULL = 0.41 mm mm$^{-1}$. APBs 77, 115, 118, and 137 did not show similar behaviors to each other, while both APB 77 and much more APB 137 displayed a remarkable decrease in the soil moisture in the first 25 cm of the soil profile (Figure 6).

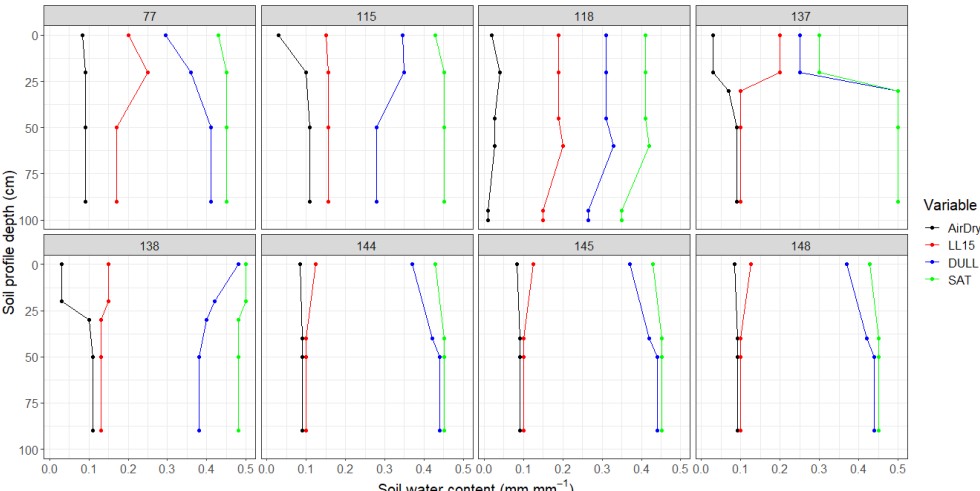

**Figure 6.** Soil moisture curves of selected APBs. Variables analyzed: soil air dry (AirDry, black line), 15 Bar (−1500 kPa) lower limit (LL15, red line), drained upper limit (DULL, blue line), and soil water saturation (SAT, green line).

### 3.3. Climate Variables

The monthly precipitation regime in the selected APBs is characterized by seven months of high precipitation (May–November), three months of low precipitation (January, February, and March), and two transitory months (April and December) (Figure 7).

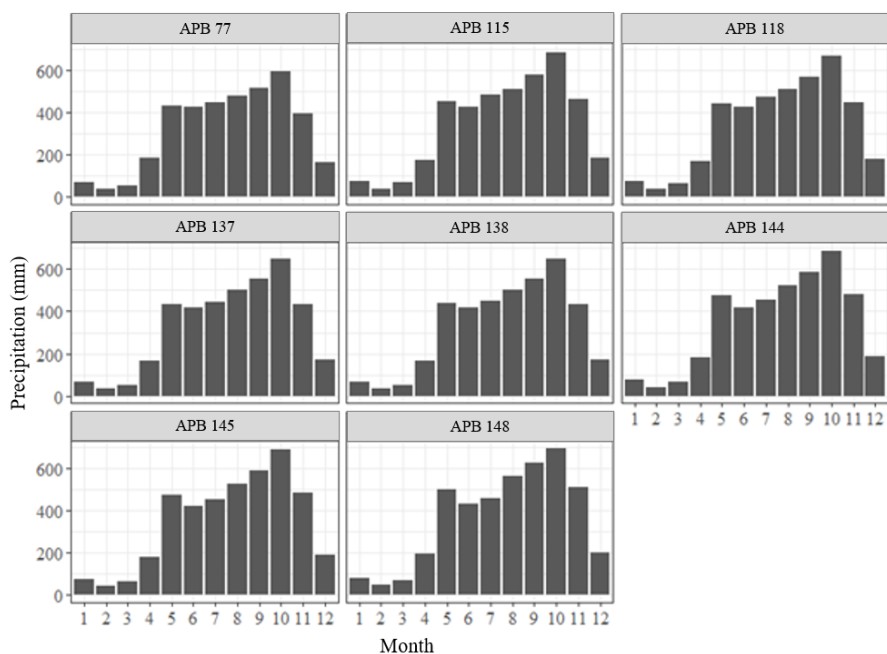

**Figure 7.** Monthly precipitation regime in each selected APBs from 1995 to 2015.

The average annual accumulated rainfall of the selected APBs is between 4537 mm and 3999 mm, with APB 148 receiving the highest annual accumulated rainfall on average (4537 mm yr$^{-1}$) and APB 77 receiving the least (3999 mm yr$^{-1}$).

Regarding temperatures in the selected APBs, maximum temperatures were found to occur in March with a mean of 32.47 °C and minimum temperatures in January with a mean of 21.90 °C, where the annual average is 26.92 °C.

Radiation is higher from January to April, with March being the month with the highest incidence, reaching values of 19.80 MJ m$^{-2}$ day$^{-1}$. The remaining months show a relatively homogeneous behavior (Figure 8).



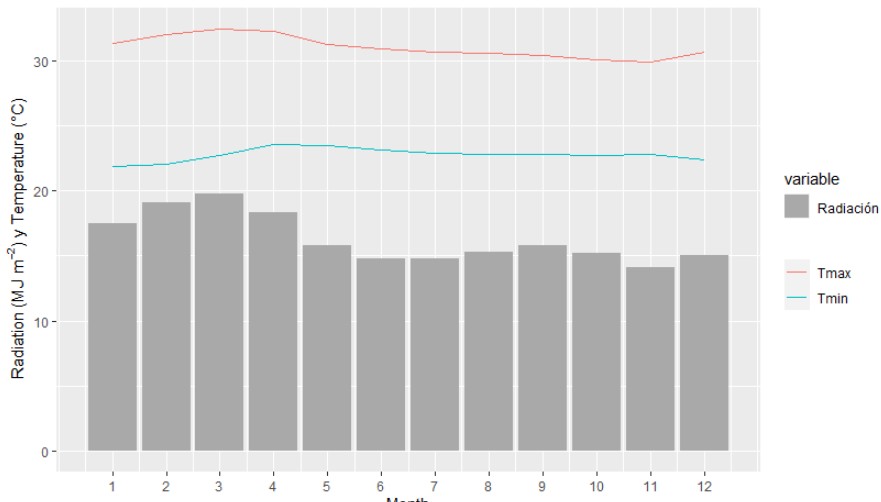

**Figure 8.** Monthly temperature and radiation regime in the selected APBs in the period from 1995 to 2015.

*3.4. Climate Change Scenarios (CCS)*

Figure 9 shows the variation among the climatological variables for the three simulations: baseline (2000–2019), CCS1 (2040–2059), and CCS2 (2080–2099).

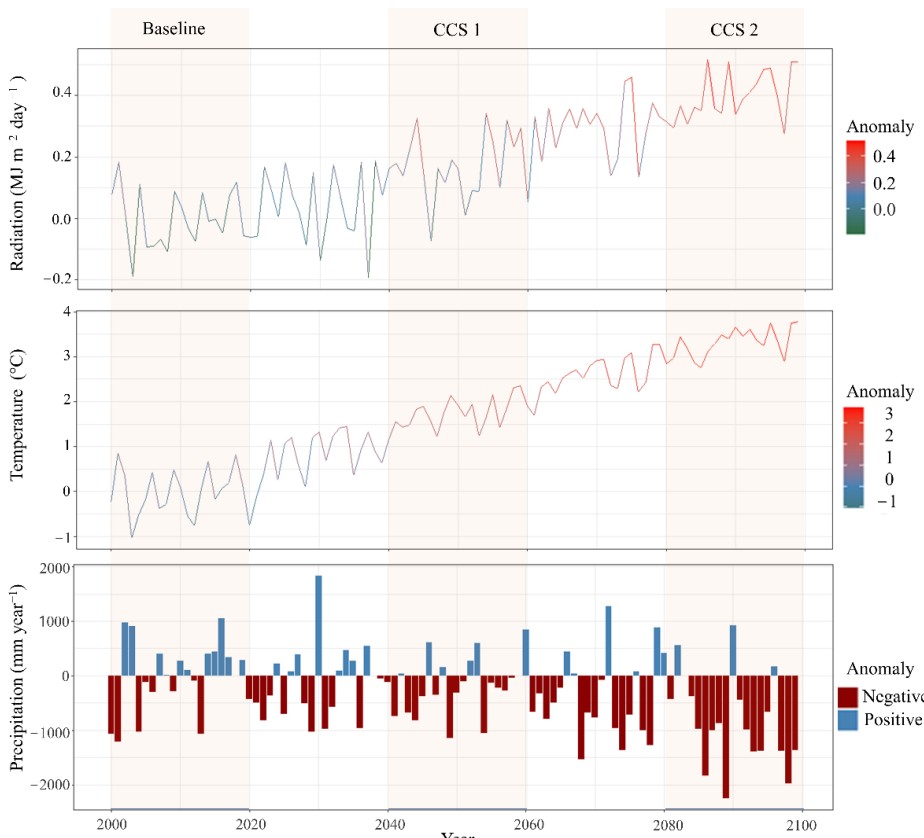

**Figure 9.** Alteration of the climatological variables of the CCS concerning the Baseline. The three assessment scenarios in the 21st century are shown in yellow boxes (baseline (2000–2019), CCS1 (2040–2059), and CCS2 (2080–2099)). For radiation and temperature, the red scale rises as the anomaly increases compared to the baseline, while precipitation is negative (red) when is lower than the baseline.

When comparing the baseline with the CCSs (Table 2), a decrease in annual cumulative precipitation of 5.55% was identified in CCS1, while the average temperature and

radiation increased by 1.73 °C and 0.18 MJ m$^{-2}$ day$^{-1}$ respectively. In the case of CCS2, greater changes are evident, with a reduction in precipitation of 18.06% and an increase in temperature of 3.31 °C and radiation of 0.40 MJ m$^{-2}$ day$^{-1}$, compared with the baseline.

**Table 2.** Comparison of climatological variables for each scenario.

| Scenario | Period | Rainfall Annual Cumulative (mm) | Temperature (°C) | | | Radiation (MJ m$^{-2}$day$^{-1}$) |
|---|---|---|---|---|---|---|
| | | | Maximum | Minimum | Mean | |
| Baseline | 2000–2019 | 4224.02 | 31.15 | 23.00 | 27.07 | 16.20 |
| CCS | 2040–2059 | 3989.67 | 32.87 | 24.74 | 28.81 | 16.38 |
| | 2080–2099 | 3461.33 | 34.46 | 26.32 | 30.39 | 16.60 |

*3.5. APSIM Model Calibration*

The results of the calibration of the models created for each of the selected APBs obtained evaluation coefficients of NSE between 0.64 and 0.74, and R$^2$ between 0.65 and 0.75, with a *p*-value less than $1.3 \times 10^{-5}$. The average error coefficients are for RMSE of 5.86 t ha$^{-1}$ (tons per hectare) and MAE of 4.90 t ha$^{-1}$ (Figure 10).

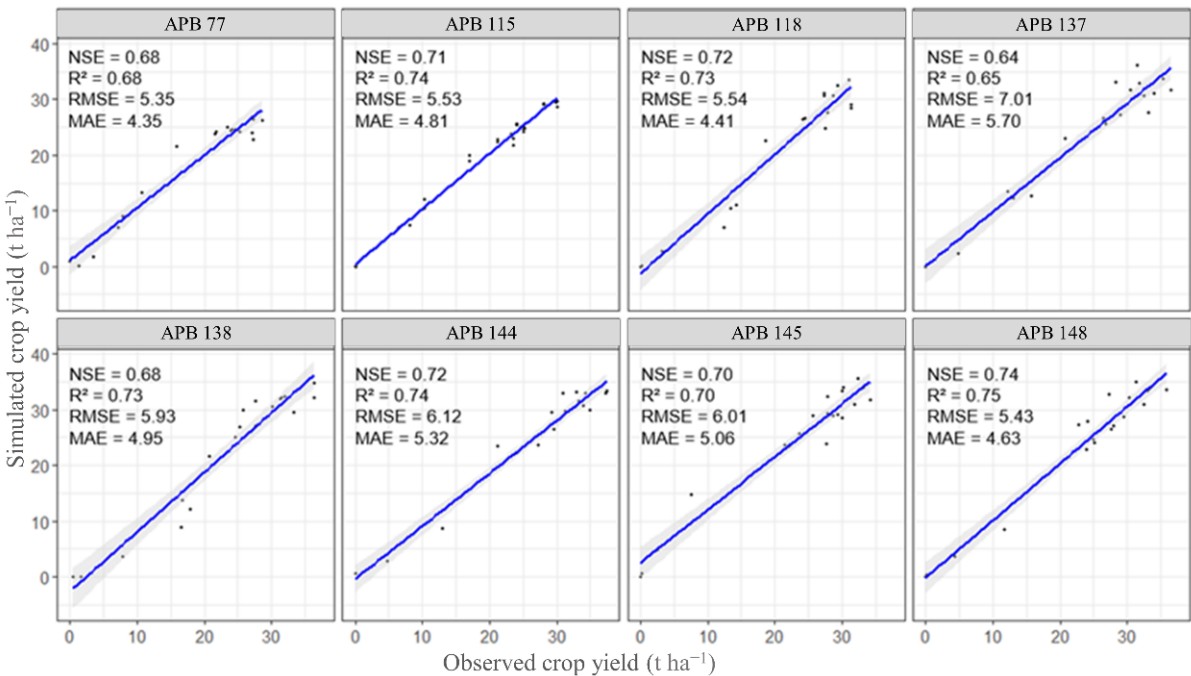

**Figure 10.** Observed versus simulated performance of APSIM models for each selected APBs for the period 1995–2015. The black dots are the data of crop yield (observed and simulated) for each selected APBs, the blue line is the best-fitting line between the data.

The simulation of APB 137 obtained the highest error coefficients (RMSE = 7.01 t ha$^{-1}$, MAE = 5.70 t ha$^{-1}$) and the lowest NSE and R$^2$ coefficients. Meanwhile, the simulation in APB 148 showed the best results according to both error and efficiency coefficients (NSE = 0.74, R$^2$ = 0.75, RMSE = 5.43 t ha$^{-1}$, MAE = 4.63 t ha$^{-1}$).

*3.6. Effects of Climate Change on Oil Palm Crop Production*

For each set of simulations (baseline, CCS1, and CCS2) of the eight selected APBs, a range of expected responses by year was generated (Figure 11). For the baseline simulations, the average yield was 21.63 t ha$^{-1}$ yr$^{-1}$ calculated for the entire planting period. It was determined that in both scenarios there was a decrease in average yield. For CCS1, the decrease was 1.7 t ha$^{-1}$ yr$^{-1}$ and for CCS2 it was 8.19 t ha$^{-1}$ yr$^{-1}$.

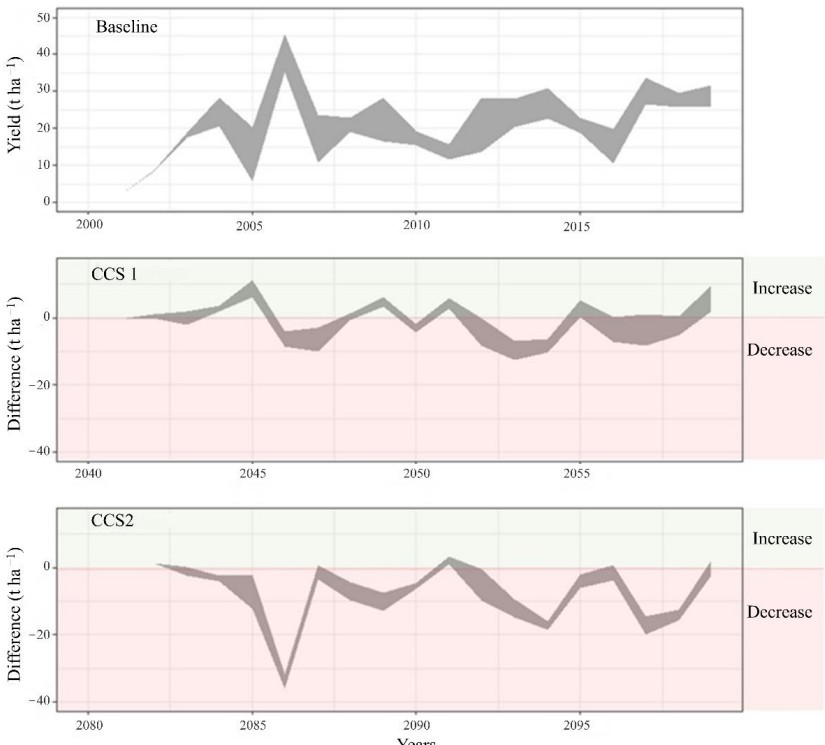

**Figure 11.** Effect of climate change on oil palm yields. For each scenario, the range of the yield performance of the eight selected APBs is presented (gray graph). When the yield (tons per hectare) in the CCS is lower than the baseline, there is a decrease (red box), otherwise there is an increase in yield (green box).

### 3.7. Oil Palm Water Requirement under the CCSs

Once it was determined that oil palm plantation yields are affected by climate change, simulations were run again, this time using APSIM's irrigation module to determine if irrigation could be used as an adaptation measure, so that yields in the CCSs would recover and at least equal those observed in the Baseline. The evaluation was conducted by determining the monthly averages of irrigation water for each of the selected APBs and each CCS.

In the case of CCS1, irrigation should be executed in January, February, March, and April with average irrigation application of 3.47, 3.39, 3.65, and 2.01 mm per day, respectively. In CCS2, the water requirements are higher, with irrigation applications in months from December to April. March is the month with the highest water requirements, with an average application of 4.76 mm per day, and even including maximum values of 7.71 mm per day (Figure 12).

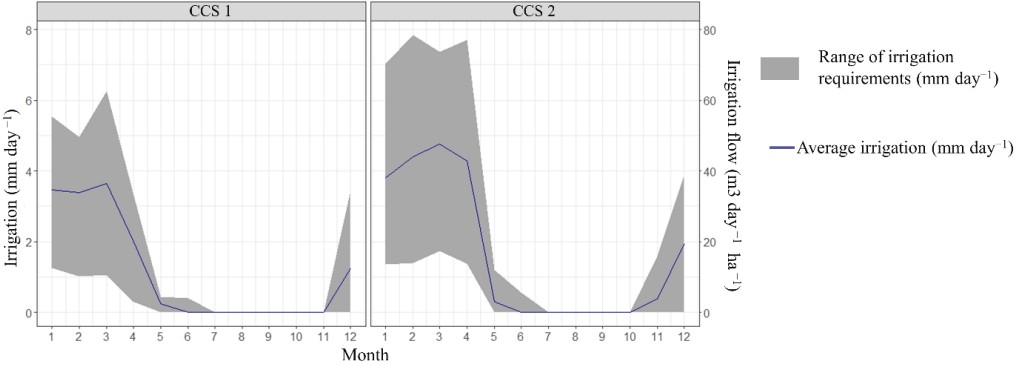

**Figure 12.** Water requirements for the selected APBs under CCSs. The range of irrigation requirements in the selected APBs is presented in mm per day (left axis) and in flow values in cubic meters per day per hectare (right axis), for each month in each CCS.

## 4. Discussion

As in this study, Alvarado Gamboa [39] showed precipitation deficits and temperature increases for the Central Pacific region of Costa Rica in the RCP 2.6 and RCP 8.5 scenarios, which will become more drastic as the century progresses, as well as an intensification of the heat wave for the CCS2 period (2080–2099).

The predominant textures of the selected APBs are sandy clay loam and clay loam, except for APB 148 which is clay, and APB 145 which has loam textures within its soil profile. The ranges of soil bulk density obtained in the laboratory generally agree with the theoretical values expressed by some authors between 1.00 and 1.4 g cm$^{-3}$ [40]. However, APB 77 and 137 in the shallowest layers (clay loam texture) have bulk density values considered to be high at 1.59 and 1.66 g cm$^{-3}$ respectively [41]. This could suggest that there is some factor, such as soil compaction, which decreases the porosity of the soil, affecting the bulk density and generating the decrease in field capacity values of the soil stratum between 0 to 25 cm depth, which is 0.295 mm mm$^{-1}$ for APB 77, and 0.25 mm mm$^{-1}$ for APB 137, which are relatively lower than the rest of the profile [42].

The calibration of the APSIM models was achieved in the eight selected APBs, obtaining acceptable efficiency coefficients and errors, consistent with the results of Huth et al. [14]. On average, the efficiency and error coefficients obtained by the models were NSE of 0.699, R$^2$ of 0.716, RMSE of 5.865 t ha$^{-1}$, and MAE of 4.903 t ha$^{-1}$ while those obtained by the developers of the oil palm module in APSIM were NSE of 0.560, R$^2$ of 0.620, and MAE of 3.490 t ha$^{-1}$ [14]. This allows giving reliability on the input variables used (precipitation, radiation, maxT, minT, BD, AirDry, LL15, DULL, ASW, and SAT) and on the response that the crop will have according to the CCSs used.

Due to the effects of climate change, a decrease in oil palm production yields is expected, with a reduction in the production of 7.86% for the CCS1 period and 37.86% for CCS2, compared to the average production of the baseline (21.63 t ha$^{-1}$ yr$^{-1}$). Other studies in Malaysia have shown that increases in temperature between 1 and 4 °C will produce a decline in yield of between 10% and 41% [43] similar to the percentage decrease in yield obtained in our study.

In both CCSs, there is an increase in the water requirements of the oil palm crop during the dry season. As the CCS becomes more intense (lower precipitation and increased temperature), the available water in the soil decreases, increasing the irrigation requirements of the oil palm [44].

From the simulations in APSIM under the CCSs, it was established that irrigation works as a solution to mitigate the effects of climate change on oil palm cultivation [20], which in our case requires an average irrigation of 3.13 mm day$^{-1}$ for CCS1 and 4.31 mm day$^{-1}$ for CCS2 (January to April). However, both economic and environmental viability should be analyzed, since climate change at the basin level will cause a decrease in water supply [45] and an increase in water demand [46]. This could result in the flows required for irrigation not being available in surface water flows. Added to this is the economic aspect, due to the costs of exploiting water for irrigation.

## 5. Conclusions

Other studies have established that climate change will produce unfavorable conditions in the oil palm crop that will affect its productivity [43,47–50].

The effects of climate change in the Central Pacific of Costa Rica for the remaining part of the 21st century will be reflected by a 5.55% decrease in the annual accumulated rainfall and an average temperature increase of 1.73 °C by mid-century (CCS1), reaching up to 18.06% in rainfall reduction and a temperature increase of up to 3.31 °C by the end of the century (CCS2). This will generate a decrease in crop production yield in the CCS1 period of 7.86% and for CCS2 of 37.86% concerning the average production of the baseline (21.63 t ha$^{-1}$ yr$^{-1}$).

The contribution of the study is vital for the logistics of the production systems for oil palm in the Central Pacific of Costa Rica since in all the analyzed CCS the hydric demand

increases, which represents a threat to the future yield of the production. Therefore, the results obtained contribute to the achievement of SDG 13, specifically in strengthening resilience and adaptive capacities to climate-related risks.

Even though irrigation was shown to be an acceptable adaptive measure to maintain current crop yields, this measure may not be environmentally or economically feasible due to the pressure that will be exerted on the water resource as climate change intensifies.

**Author Contributions:** Conceptualization, F.W.-H.; methodology, F.W.-H., V.S.-N., N.G.-C. and R.P.d.S.; software, F.W.-H.; validation, F.W.-H.; formal analysis, F.W.-H., V.S.-N., N.G.-C. and R.P.d.S.; investigation, F.W.-H.; resources, F.W.-H. and N.G.-C.; data curation, F.W.-H.; writing—original draft preparation, F.W.-H.; writing—review and editing, F.W.-H., V.S.-N., N.G.-C. and R.P.d.S.; visualization, F.W.-H.; supervision, F.W.-H. and N.G.-C.; project administration, F.W.-H.; funding acquisition, F.W.-H. All authors have read and agreed to the published version of the manuscript.

**Funding:** This research received no external funding.

**Data Availability Statement:** Not applicable.

**Acknowledgments:** The following people contributed to this research; Rafaela Hernández Ortega, María Laura Monge Espinach, Isabel Guzmán Arias, Hugo Hidalgo León, Karolina Villagra Mendoza, Marvin Villalobos Araya, Johan Fernández, Ronald Fernández, Randall Barquero Solano.

**Conflicts of Interest:** The authors declare no conflict of interest.

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
