# Peer review of "Quantification and Evaluation of Water Requirements of Oil Palm Cultivation for Different Climate Change Scenarios in the Central Pacific of Costa Rica Using APSIM"

_agronomy, doi:10.3390/agronomy13010019_

Round 1

Reviewer 1 Report

This manuscript (title: Quantification and evaluation of water requirements of oil palm cultivation for different climate change scenarios in the Central Pacific of Costa Rica using APSIM) assessed the future impacts of climate change on oil palm productions in the Central Pacific of Costa Rica by APSIM model. The research fits the topics of Agronomy. In general, this study was conducted with clear objectives and suitable methods. However, as the citations in the Introduction section, there have been a lot of studies on this topic, which caused this study with lack of novelty. I suggested the authors reorganized the “Introduction” and “Discussion” sections. I believed that the manuscript could be considered for publication after addressing a few technical concerns. Besides, current version suffers a number of weaknesses covering dreary analyzing of the results and clarification confusions.

1. In the “Introduction” section, the importance of oil palm production in the studied areas was not given, which makes a weaker significance of this study.

2. L64: what is “an experimental design”?

3. L85: The root depth of 90 cm may not be enough for oil palm trees.

4. L89: Please clarify the periods of the average climate variables values.

5. L97: Why did not you use the SSP or RCP scenarios?

6. There are too much contents and figures about the changes of climate variables. However, based on the title, the main contents should be the impacts of climate change on oil palm and irrigation adaptations, which only has a very small parts in the “Results” and “Conclusions” sections.

Because I am not native English speaker, I did not pay much attention on the English statement. However, I do suggest that authors should find native English writer to improve the sentences and paragraphs.

Author Response

Dear Reviewer 1

We are very grateful for your comments and suggestions. We now proceed to indicate the corrections and improvements made by the researchers.

  1. In the “Introduction” section, the importance of oil palm production in the studied areas was not given, which makes a weaker significance of this study.

The importance of oil palm production in Costa Rica and the Central Pacific zone (where the study area is located) was highlighted in the "Introduction".

  1. L64: what is “an experimental design”?

The line indicating that the study was carried out using an experimental design (first paragraph of "Materials and Methods") was deleted to avoid misunderstandings among the readers of the manuscript. 

  1. L85: The root depth of 90 cm may not be enough for oil palm trees.

The 90 cm root depth (in section “2.2. Soil physical parameters”) was justified by literature showing that the greatest oil palm root density is found at depths of 20 cm to 60 cm.

  1. L89: Please clarify the periods of the average climate variables values.

It was rearranged and clarified in the manuscript section "2.3. Climate data" that the averages of the climate variables evaluated are from the study period 1995 to 2015.

  1. L97: Why did not you use the SSP or RCP scenarios?

In section "2.4. Climate Change Scenarios (CCS)" it was added that the CCS used in this paper was based on the study by Hidalgo et al. (2017) (doi: 10.1007/s10584-016-1786-y), in which 107 climate runs of 48 general circulation models (GCMs) of the Coupled Model Intercomparison Project 5 (CMIP5), which is a model that considers SSP-RCP scenarios.

  1. There are too much contents and figures about the changes of climate variables. However, based on the title, the main contents should be the impacts of climate change on oil palm and irrigation adaptations, which only has a very small parts in the “Results” and “Conclusions” sections.

The results obtained in sections 3.2 to 3.5 are the basis for determining the water requirements of the oil palm crop under the climate change scenarios described in section 3.7. These results are analyzed in the “Discussion” section and highlighted in the “Conclusions”. 

On the other hand, the novelty of the study is added in the Conclusions.

Reviewer 2 Report

The research paper has a good scientific level. The content of the paper is valuable from the theoretical and especially from the practical point of view. The article presents a current topic. The aim of research is interesting and beneficial for this research field. The topic is convenient for the scope of the journal. The title of scientific article is clear and it sufficiently reflects content. The abstract and key words are informative. The figures appropriately complement the presentation of the scientific work results. The tables are acceptable in this state. The manuscript is well organized. In the manuscript are some irregularities which need corrections.

1. Keywords should start with lowercase letters.

2. Why was APSIM used to simulate the effect of climate change on crops? What are the main advantages of the used calculation tool?

3. In the INTRODUCTION, I recommend stating in which research APSIM has already been used for climate change prediction.

4. (Chapter 2.2) Water transport cannot be separated from heat transport. For the stated reasons, it would be appropriate to state how the individual parameters of the soil change depending on the moisture content for the specified location. What was the thermal and electrical conductivity of the soil? What other physical parameters were investigated to obtain comprehensive knowledge about soil properties?

5. Line 85: Pressure (15 Bar) must be given in SI units.

6. (Chapter 2.3) Please add a more precise description of the climate data collection and method of climate data evaluation.

7. The writing of numerical values with a gap before the unit of the physical quantity must be checked throughout the text.

8. Equations 2 and 3 do not need to be stated, they are generally known.

9. What is the reliability of the implemented computational simulation?

10. The CONCLUSION should be extended. I recommend clearly emphasizing the novelty of the research in Conclusion.

GENERAL JUDGEMENT

The paper is acceptable for publication after minor revision.

Author Response

Dear Reviewer 2

We are very grateful for your comments and suggestions. We now proceed to indicate the corrections and improvements made by the researchers.

  1. Keywords should start with lowercase letters.

The key words were corrected and written in lowercase letters.

  1. Why was APSIM used to simulate the effect of climate change on crops? What are the main advantages of the used calculation tool?

In the "Introduction" we described some of the advantages of using APSIM, which is an excellent agricultural systems simulator, with a variety of modules that can be used, including the specific module for oil palm cultivation, calibrated, and validated for tropical conditions, in addition to the inclusion of climate data and projections is relatively simple, and allows the use of an irrigation module, fulfilling the objectives of this study.

  1. In the INTRODUCTION, I recommend stating in which research APSIM has already been used for climate change prediction.

Literature on the use of APSIM to evaluate crop yields under climate change scenarios was added in the "Introduction".

  1. (Chapter 2.2) Water transport cannot be separated from heat transport. For the stated reasons, it would be appropriate to state how the individual parameters of the soil change depending on the moisture content for the specified location. What was the thermal and electrical conductivity of the soil? What other physical parameters were investigated to obtain comprehensive knowledge about soil properties?

Heat transport, thermal and electrical conductivity were not evaluated in this study because these are not soil parameters required in the APSIM soil module, nor part of the objectives of the study.

The physical parameters analyzed in the study were those used for the soil model: texture, bulk density, soil air dry, 15 Bar (-1500 kPa) lower limit, drained upper limit, available soil water, and soil water saturation.

However, the characterization of the soil order of the study area was added in section "2.2. Soil physical parameters", to give a better context about the soil of the APBs.

  1. Line 85: Pressure (15 Bar) must be given in SI units.

The value of 15 bar in SI units, which corresponds to 1500 kPa, has been added in section "2.2. Soil physical parameters".

  1. (Chapter 2.3) Please add a more precise description of the climate data collection and method of climate data evaluation.

A detailed description of the climate data collection and method of climate data evaluation was added in section “2.3. Climate data”

  1. The writing of numerical values with a gap before the unit of the physical quantity must be checked throughout the text.

The manuscript was checked and verified that there was a space between the numerical values and the units of their magnitude.

  1. Equations 2 and 3 do not need to be stated, they are generally known.

Equations 1 and 4 as well as 2 and 3 were considered important and contributed to the understanding of the statistical evaluation of the study, so it was decided to keep all the equations to preserve consistency in the manuscript.

  1. What is the reliability of the implemented computational simulation?

The reliability of the computational model used is supported by the results obtained from the statistical coefficients that evaluated it, reaching values of NSE of 0.699, R2 of 0.716, RMSE of 5.865 t ha-1, and MAE of 4.903 t ha-1, which when compared with those obtained in the oil palm module in APSIM (NSE of 0.560, R2 of 0.620 and MAE of 3.490 t ha-1), are higher. This indicates that the calibrated model used has a good prediction performance.

  1. The CONCLUSION should be extended. I recommend clearly emphasizing the novelty of the research in Conclusion.

The novelty of the study was added in the "Conclusions".

Reviewer 3 Report

The study is devoted to modeling the yield of oil palm in the conditions of climate change in Costa Rica. Special attention is paid to the precipitation regime, since it is a yield limiting factor; temperature and radiation are also taken into account. The cesm1_cam5 scenario of climate change was used; the yield is predicted using the APSIM-Oil Palm. The choice of the model, the data for the model, the generation of daily meteorological data and the filling of gaps, the selection of 8 experimental points are logically substantiated. The conclusions are justified. In the XXI century, a decrease in precipitation and yield of oil palm is predicted.

There are several unclear passages in the text.

Fig.1: the values of seconds 0.000” in all coordinates are unclear, ‘for example. -84°6’0.000”. Why are they all equal to zero? What for the decimal point?

It is not clear from Fig. 1 how far the rain gauging stations are from the experiment site, since there is no coordinate grid step in the small figure. Maybe we need the exact coordinates of the stations.

Lines 128-130: "During calibration, models were generated for each of the selected APBs to achieve NSE => 0.6 and R2 => 0.65 in accordance with the values obtained by the creators of the module for the oil palm [25]." In the [25], Nuth et al 2014, paragraph 3.2:"from NSE more than 0.5 are satisfactory, and values greater than 0.65 indicate good or very good characteristics of the model." Why did the authors take NSE more than 0.6? Where is the boundary R2>0.65 taken from?

Figure 3: “Daily climate observations in 2000 – 20015”- extra zero.

Lines 162-163: " the cluster with the highest scores was a cluster of 8 followed by 2 " - what are clusters 8 and 2? Probably need a link to the Fig. 4 and the cluster numbers in the picture.

Fig. 4 – "Clusters of oil palm plantations by monthly yield" - – what is monthly yield?

Lines 172-173 "APB 72 was pre-selected, but was replaced by APB because it better reflects the average behavior of the analyzed population (Figure 5)" - The selection criterion is diversity, what does the average behavior have to do with it?

“Table 2. Comparación de las variables climatológicas para cada escenario” – need an English translation. Probably the unit of measurement of radiation is MJ m-2 day-1.

Best regards

Author Response

Dear Reviewer 3

We are very grateful for your comments and suggestions. We now proceed to indicate the corrections and improvements made by the researchers.

  1. Fig.1: the values of seconds 0.000” in all coordinates are unclear, ‘for example. -84°6’0.000”. Why are they all equal to zero? What for the decimal point?

The values of the seconds in the coordinates are shown as "0.000" because the coordinate grid created had an accuracy of 3 decimal places. Figure 1 was modified with a grid with a smaller spacing and zero decimals in order to display the seconds values with fewer decimal places.

  1. It is not clear from Fig. 1 how far the rain gauging stations are from the experiment site, since there is no coordinate grid step in the small figure. Maybe we need the exact coordinates of the stations.

Figure 1 was modified with a smaller grid spacing to improve the visualization and identification of the location of the rain gauging stations. In addition, a scale was added to the map to allow the measurement of distances.

  1. Lines 128-130: "During calibration, models were generated for each of the selected APBs to achieve NSE => 0.6 and R2 => 0.65 in accordance with the values obtained by the creators of the module for the oil palm [25]." In the [25], Nuth et al 2014, paragraph 3.2:"from NSE more than 0.5 are satisfactory, and values greater than 0.65 indicate good or very good characteristics of the model." Why did the authors take NSE more than 0.6? Where is the boundary R2>0.65 taken from?

In section "2.5. APSIM simulation process" the values of NSE and R2 were corrected, to NSE => 0.56, and R2 => 0.62, which are the coefficients of the statistics obtained in the creation of the oil palm module in APSIM when predicting yield (Huth et al., 2014 (doi: 10.1016/j.envsoft.2014.06.021)). These coefficients were used during the calibration process in order to evaluate the model, expecting to obtain equal or higher values.

In the study by Huth et al. (2014) he refers to Moriasi et al. (2007) (doi: 10.13031/2013.23153), who suggests that NSE values greater than 0.5 are considered satisfactory, and those greater than 0.65 indicate good to very good performance of the prediction model. These NSE values were those used by Huth et al. to evaluate the oil palm model, but they are not the ones used in our study.

3: “Daily climate observations in 2000 – 20015”- extra zero.

Figure 3 "Daily climate observations in 2000 - 20015" was corrected to "Daily climate observations in 2000 - 2015", eliminating the extra zero.

  1. Lines 162-163: " the cluster with the highest scores was a cluster of 8 followed by 2 " - what are clusters 8 and 2? Probably need a link to the Fig. 4 and the cluster numbers in the picture.

It was clarified in the manuscript in section "3.1. Selection of study APBs" that the result of the Silhouette method and the clustering with Computers Hierarchical Clustering and Cut the Tree method obtained two possible groupings of the data: 2 and 8, of which the latter was selected because it maximizes the representation of the APBs performance behavior. The 8 clusters are shown in Figure 4 (gray squares with dotted lines).

  1. Fig. 4 – "Clusters of oil palm plantations by monthly yield" - – what is monthly yield?

The monthly yield of palm oil plantation (Figure 4) refers to the selection of the APBs of study that was made by the Silhouette method and the clustering with Computers Hierarchical Clustering and Cut the Tree method, in which the historical yield values of all the APBs of the area under study was used. The above was added in section "2.1. Study area" in the "Materials and Methods" and "3.1. Selection of study APBs" of the "Results" in the manuscript.

  1. Lines 172-173 "APB 72 was pre-selected, but was replaced by APB because it better reflects the average behavior of the analyzed population (Figure 5)" - The selection criterion is diversity, what does the average behavior have to do with it?

The paragraph was clarified, as it was implied that APB 118 was selected for its diversity, when in fact APB 118 was selected for being closer to the average yield of the APBs under study, and that is why it is used to reflect the behavior of the population.

  1. “Table 2. Comparación de las variables climatológicas para cada escenario” – need an English translation. Probably the unit of measurement of radiation is MJ m-2 day-1.

The title of Table 2 was translated and edited in the manuscript, and the units of measurement of radiation in the table were corrected to MJ m-2 day-1.
